# Effect of Water-Soluble Polymers on the Rheology and Microstructure of Polymer-Modified Geopolymer Glass-Ceramics

**DOI:** 10.3390/ma17122856

**Published:** 2024-06-11

**Authors:** John M. Migliore, Patrick Hewitt, Theo J. Dingemans, Davide L. Simone, William Jacob Monzel

**Affiliations:** 1Department of Applied Physical Sciences, The University of North Carolina at Chapel Hill, Chapel Hill, NC 27599, USA; jmigs@email.unc.edu (J.M.M.); tjd@unc.edu (T.J.D.); 2Materials and Manufacturing Directorate, Air Force Research Laboratory, AFRL/RXNP, Dayton, OH 45324, USA; patrick.hewitt.ctr@us.af.mil (P.H.); davide.simone@us.af.mil (D.L.S.); 3UES, Inc. A BlueHalo Company, Dayton, OH 45432, USA

**Keywords:** geopolymer, polymer-modified, composites, glass-ceramics, Sol-gel processes, microstructure-prefiring, porosity, rheology, hybrid, high-performance

## Abstract

This work explores the effects of rigid (0.1, 0.25, and 0.5 wt. %) and semi-flexible (0.5, 1.0, and 2.5 wt. %) all-aromatic polyelectrolyte reinforcements as rheological and morphological modifiers for preparing phosphate geopolymer glass–ceramic composites. Polymer-modified aluminosilicate–phosphate geopolymer resins were prepared by high-shear mixing of a metakaolin powder with 9M phosphoric acid and two all-aromatic, sulfonated polyamides. Polymer loadings between 0.5–2.5 wt. % exhibited gel-like behavior and an increase in the modulus of the geopolymer resin as a function of polymer concentration. The incorporation of a 0.5 wt. % rigid polymer resulted in a three-fold increase in viscosity relative to the control phosphate geopolymer resin. Hardening, dehydration, and crystallization of the geopolymer resins to glass-ceramics was achieved through mold casting, curing at 80 °C for 24 h, and a final heat treatment up to 260 °C. Scanning electron microscopy revealed a decrease in microstructure porosity in the range of 0.78 μm to 0.31 μm for geopolymer plaques containing loadings of 0.5 wt. % rigid polymer. Nano-porosity values of the composites were measured between 10–40 nm using nitrogen adsorption (Brunauer–Emmett–Teller method) and transmission electron microscopy. Nanoindentation studies revealed geopolymer composites with Young’s modulus values of 15–24 GPa and hardness values of 1–2 GPa, suggesting an increase in modulus and hardness with polymer incorporation. Additional structural and chemical analyses were performed via thermal gravimetric analysis, Fourier transform infrared radiation, X-ray diffraction, and energy dispersive spectroscopy. This work provides a fundamental understanding of the processing, microstructure, and mechanical behavior of water-soluble, high-performance polyelectrolyte-reinforced geopolymer composites.

## 1. Introduction

Geopolymers represent a class of glass-ceramics with extreme heat resistance, low shrinkage, and high compressive strength relative to Portland cement [1,2,3]. Although not commonly equated in the literature to sol-gel systems, these geopolymer materials may be considered a variant of acid- or base-catalyzed sol-gel chemistry, with solid oxide powder as the starting material instead of monomeric precursors. Aluminosilicates are the most common geopolymer precursors but other elements may be used. For example, phosphate geopolymers (PGPs) can be produced with any divalent or trivalent oxide [4]. Aluminosilicate materials, such as kaolin, are mined and processed to form relatively pure and reactive materials, while less pure SiO_2_/Al_2_O_3_ materials are commonly sourced from low-cost industrial waste by-products [5,6,7,8,9,10,11]. The hydrolytic (aqueous-based) sol-gel process to create geopolymer materials typically utilizes a mixture of oxides, such as metakaolin and fly ash, strongly acidic or alkaline reactants, and water [12,13,14].

Geopolymers fall into two categories: alkaline (e.g., sodium-, potassium-, or cesium-based) and acidic (e.g., phosphorous-based). The chemical process in Figure 1a depicts the dissolution and geopolymerization of an aluminosilicate (metakaolin) particulate in aqueous phosphoric acid. The acidic conditions favor the dissolution of Al species as divalent and trivalent oxides to form stable phosphates; thus, Si-O-P networks are unstable [4]. Further dissolution results in a phase-separated mixture of Si-rich and Al/P-rich regions [15,16,17,18]. Formation of geopolymer plaques from acid catalysts often results in domain sizes < 100 nm [19,20,21]. The covalent network formed during geopolymerization follows the general mechanism provided in Figure 1b: (1) solvation/dissolution of the aluminosilicate starting material, (2) formation and polymerization of alumina/silica oligomers into amorphous hydrates, and (3) final polycondensation of alumina/silica species into a 3D amorphous–crystalline network [22,23]. Alkaline geopolymer systems form via the nucleation of highly condensed, spherical colloidal nanoparticles that connect to form a network, while acid-catalyzed geopolymers provide faster hydrolysis and slower polycondensation, leading to the formation of a polymeric network that then densifies [24,25].

A range of synthesis and processing conditions result in variations in the microstructure and mechanical properties of geopolymers. The formation of geopolymer materials from alkaline sources is well-understood [9,10,27,28,29,30]. In contrast, acidic activation with phosphoric acid has garnered additional attention in recent years due to the increased mechanical strength, thermal stability, and adsorption capabilities of PGPs [31,32]. Perera et al. reached a maximum compressive strength of ~146 MPa for a PGP system with a 1:1 Si/Al molar ratio [33]. Tchakouté et al. demonstrated a 150% increase in modulus in metakaolin-based PGPs as a function of phosphoric acid concentration [34]. Cui et al. formulated PGPs with heat resistance up to 1500 °C from a heat treatment lower than common ceramic materials [35]. Djobo et al. utilized increasing molar concentrations of phosphoric acid (6, 8, and 10 mol/L) to reduce porosity from 22–10% and maintain thermal stability up to 1000 °C [36]. Additional reviews regarding the variations in synthetic conditions of alkaline and acidic geopolymers are provided by numerous authors [20,26,37,38,39,40,41].

Thus, geopolymers demonstrate an economical and environmentally sustainable material for applications in fire- and corrosion-resistant coatings, concrete reinforcement, waste-water treatment, and high-performance composites [42,43,44,45,46,47]. However, applications of geopolymers are limited as virgin SiO_2_/Al_2_O_3_ ceramic materials display low fracture toughness, low ductility, and brittle mechanical properties [48,49,50,51]. Researchers have explored a range of geopolymer additives to combat the microstructural defects that emerge during the curing process and improve mechanical properties [52,53,54,55,56]. Mechanical reinforcements, such as fibers and organic polymers, have been used to improve the toughness of cement and geopolymer materials [57,58,59,60,61,62,63,64,65]. Fibers, whether chopped or continuous, serve as excellent bulk reinforcement materials but act primarily at the micro-scale and have limited impact on resin morphology [66,67]. Organic polymers act as additives across different length scales and domain sizes to improve compatibility between the ceramic matrix and other reinforcing agents. Compatibilization can be achieved through class I (non-covalent, weak bonding) and class II (covalent, strong bonding) network formations [68]. However, most polymers are not chemically compatible with aqueous processing, which leads to phase separation, large domains, chemical degradation of the polymer, and insufficient interface/interphase strengths [69,70,71,72].

The use of organic polymers to generate organic/inorganic geopolymer composites and hybrid materials has garnered much attention in recent years [7,37,48,73]. These polymers are mostly commodity polymers such as epoxies, polyesters, siloxanes, carboxymethyl cellulose, polyvinyl acetates, acrylic acid butyl acrylates, and polyvinyl chlorides [43,45,74,75,76,77]. Zhang et al. reported the homogeneous mixing of water-soluble polymers, polyacrylic acid, sodium polyacrylate, polyethylene glycol, polyvinyl alcohol, and polyacrylamide during sol-gel processing [54]. Chen et al. studied the effects of polyacrylate in a geopolymer composite to reveal toughening of the polymer-modified system at SiO_2_/Al_2_O_3_ ratios below 2.5 [2]. Chen et al. also demonstrated a reduction in pore size of <10 nm for slag-based geopolymers with the incorporation of less than 1.0 wt. % sodium polyacrylate [78]. Rondinella et al. utilized <5.0 wt. % Chitosan, a naturally derived polysaccharide, to reduce water absorption (open porosity) by 33.0% in an alkali geopolymer, which enabled an improvement in fracture toughness of up to 90% [79]. Roviello et al. synthesized type I (non-covalent) and type II (covalent) geopolymer hybrid materials with commercial oligomeric dimethylsiloxane and fly-ash-based starting materials [68]. Glad et al. utilized the alkoxysilane functionalization of methacrylate groups to increase Weibull modulus by ~50% and reduce mesoporosity by a factor of 10 [51].

The above studies offer compelling evidence that PGP + Polymer composite materials warrant further investigation. However, questions still remain in regard to (1) how the polymer additive directs the formation of the geopolymer microstructure and (2) what domain sizes and interfacial strengths result in improved toughening for these organic/inorganic hybrid systems.

Additionally, the use of commodity polymers severely limits the high-temperature application of these materials [10,48]. While porosity refinement can result in improved mechanical properties even after polymer degradation, this degradation at elevated temperatures limits ductile phase toughening mechanisms, such as pinning and crack bridging, and contributes to weight loss and crack formation. The use of more thermally stable polymers may allow for improved properties at >300 °C and decouple the weight loss regimes between water and organic polymer.

To the best of our knowledge, few studies have incorporated high-performance polymers as reinforcement agents in PGP systems. High-performance polymers, such as all-aromatic polyamides and polyimides, provide an avenue toward improvements in fracture toughness, tensile strength, and thermal stability [80,81,82,83,84,85,86]. Additionally, the utilization of water-soluble, high-performance polymers eliminates the need for harmful solvents that are associated with non-aqueous composite material processing [87,88]. Thus, work focused on the formulation, processing, and characterization of water-soluble, high-performance PGP + Polymer composites and will further expand the potential applications of these materials.

Water-soluble, high-performance polyelectrolytes have received much attention [89,90,91,92,93]. The linear, all-aromatic polyamide poly(2,2′-disulfonyl-4,4′-benzidine terephthalamide) (PBDT) and kinked, all-aromatic polyamide poly(2,2′-disulfonyl-4,4′-benzidine isophthalamide) (PBDI) represent a class of high-performance polymers with liquid crystal and hydrogel properties [94,95,96,97,98]. PBDT is referred to as rigid due to the all-*para* substitutions of the aromatic rings, while PBDI is referred to as semi-flexible from the isophthaloyl *meta*-substitution as displayed in Figure 2. These materials have been shown to exhibit exceptional tensile strength as a nanocomposite polymer matrix and provide excellent thermal stability at temperatures above 350 °C [91,96]. Previous work has demonstrated that polymer reinforcement of geopolymer materials improves the bulk properties by controlling the microstructure and reducing crack initiation/growth [55,99]. Thus, we hypothesize that the incorporation of PBDT and PBDI will reduce microporosity and serve as reinforcements within the geopolymer particulate and binding phases. Water-soluble, high-performance polymer reinforcement will improve strength and toughness while maintaining high processability and thermal stability.

Herein, we present the synthesis and characterization of a class I PGP composite material reinforced with high-performance, water-soluble polyelectrolytes. The incorporation of an organic polymer during the sol-gel synthesis revealed significant effects on the rheology and microporosity of the metakaolin-based PGP system. Little impact was made on the crystallinity of the composite material due to the class I nature of the hybrid materials. However, sufficient loadings of a rigid polymer (PBDT, 0.5 wt. %) and a semi-flexible polymer (PBDI, 2.5 wt. %) improved particle suspension/interactions within the sol-gel resin, leading to microporosity control during the curing process. The following results provide insight into the compatibility between water-soluble, high-performance polymers and their impact on the rheological, structural, and micromechanical properties of polymer-reinforced, metakaolin-based PGP composites.

## 2. Materials and Methods

### 2.1. Materials

PowerPozz^TM^ (off-white powder) was acquired from Advanced Cement Technologies and calcined at 700 °C for 2 h to ensure complete conversion to metakaolin. The calcination produced an orange-brown powder. The chemical composition of PowerPozz^TM^ can be found in Table 1. Average particle size of calcined PowerPozz^TM^ (metakaolin) was 34.6 μm and measured with a Beckman Coulter LS 13 320 laser diffraction particle size analyzer. *o*-Phosphoric acid (H_3_PO_4_, 85% *w*/*w*, Certified ACS), hydrochloric acid (HCl, 36.5–38% *w*/*w*, certified ACS plus), chloroform (CHCl_3_, certified ACS, ≥99.8% purity), ethanol (EtOH, ACS reagent, ≥99.5%), and sodium carbonate anhydrous (Na_2_CO_3_, powder, certified ACS, ≥99.5% purity) were purchased from Fisher Chemical^TM^. Acetone (C_3_H_6_O, ACS, ≥99.5% purity), methanol (MeOH, ACS, ≥99.8% purity), and sodium hydroxide (NaOH, 50% *w*/*w*) were purchased from VWR International. 2,2′-Benzidinedisulfonic acid (BDSA, powder, 20% water at maximum, ≥70% purity) was purchased from TCI America and purified 4× until the diamine monomer appeared as an off-white powder [101]. Terephthaloyl chloride (TPC, flakes, ≥99% purity) and isophthaloyl chloride (IPC, flakes, ≥99%) were purchased from Sigma-Aldrich and sublimed before use. Unless stated otherwise, all materials and solvents were used as received.

### 2.2. Preparation of Geopolymer Composite Plaques

The hydrolytic sol-gel process to form geopolymer composites included PBDT or PBDI, metakaolin, and 9M phosphoric acid. An 85 wt. % *o*-phosphoric acid solution was diluted with deionized water to generate a 9M phosphoric acid stock solution. The syntheses of PBDT and PBDI have been described elsewhere [89,101,107]. Formulations of virgin and polymer-reinforced geopolymer resins follow the same steps. Mixing quantities of the resins can be found in Table 2. The geopolymer phase was fixed at 1:1:1 Si:Al:P with 9M H_3_PO_4_ (2 SiO_2_ • Al_2_O_3_ • 2 H_3_PO_4_ • 9.6 H_2_O). A procedure outlining the synthesis of PGP + 2.5 wt. % PBDI will serve as an example in this section.

PBDI was dried under vacuum at 60 °C overnight. Dissolution of 0.704 g (1.36 mmol) PBDI in 26.9 g (0.171 mol) 9M phosphoric acid was achieved while heating and stirring (50 °C, 200 RPM) for 16 h. The resulting PBDI and 9M phosphoric acid solution was targeted to equal 2.5 wt. % (*w*/*w*). Upon dissolution, the acidic solution was transferred to a mixing container. Metakaolin (20.0 g, 87.0 mmol) was introduced to the solution in 3 parts and mixed with a planetary mixer (Flacktek Speedmixer DAC 12000-300VAC) following each addition. The sol-gel resin was mixed using a modified dispersion blade at 2000 RPM for 5 min. The dispersed solution was then transferred to the planetary mixer for 5 min at 2000 RPM to degas the resin. Resin was poured into an open face 76 × 76 × 3 mm polytetrafluoroethylene (PTFE)-lined steel mold and sealed with adhesive. Clamps were used to secure the mold. The mold was then held vertically and agitated to allow the resin to settle. Once sealed, the mold was placed in a Heratherm convection oven to begin the curing cycle. Images of the geopolymer resin and final plaque are displayed in Figure 3. The curing cycle is also provided in Figure 3 and follows 4 main parts: (1) begin at room temperature and cure to 80 °C for 24 h, (2) post-cure at 200 °C and hold for 3 h, (3) cool to 50 °C and remove the sample from the mold, and (4) place the sample in a stainless-steel bag and heat to 260 °C for 3 h. No additional thermal treatments were performed prior to analysis.

### 2.3. Characterization of Geopolymer Composite Plaques

#### 2.3.1. Rheology

Immediately following the sol-gel mixing process, rheology analysis was performed on the geopolymer resins. Samples were studied using a TA Instruments ARES-G2 with a forced convection oven under air flow. All experiments were run in quadruplicate at 25 °C. A moist Kimwipe^®^ was placed in the bottom of the oven to maintain humidity and limit water loss during long sample collection times. Parallel plates (25 mm diameter, stainless steel, disposable) were equipped to mitigate particle–geometry interactions and provide sufficient torque sensitivity across a wide viscosity range. All samples were pre-sheared at 1 s^−1^ for 120 s to eliminate differences in sample loading history. Experiments were performed using both oscillatory and steady shear procedures.

#### 2.3.2. Thermal Gravimetric Analysis (TGA)

Final incorporation of PBDT and PBDI was quantified using a TA Instruments TGA 550 equipped with a wire-wound (Pt/Rh), evolved gas analysis furnace. Samples were loaded on flame-dried TA Instruments platinum pans that were tared before each heating cycle. The geopolymer samples were crushed into a fine powder (≤10 μm) and stored under ambient conditions for 72 h before loading. Prior to the final 1000 °C temperature ramp, samples were heated from room temperature to 200 °C at 10 °C/min and held at 200 °C for 30 min to remove physiosorbed water. Samples were then equilibrated to 50 °C, heated to 1000 °C at 10 °C/min, and replotted accounting for water mass loss. The mass of all samples ranged from 25–30 mg and experiments were performed in triplicate.

#### 2.3.3. Scanning Electron Microscopy (SEM)

Sample preparation was performed on all geopolymer plaques using a surface area of ~2 cm^2^. A 400-grit silicon carbide (SiC) polishing pad was first used to remove the upper surface of the geopolymer plaque. Additional fine grit selections using nylon pads and diamond polishing pastes were made in series until the sample surface was ground to a tolerance of ~1 μm. Non-aqueous solvents were used as lubricants to avoid loss of the water-soluble polymer. Sonication was performed to remove small particles and excess carbon collected during surface removal. Samples were dried at 60 °C overnight.

Prior to SEM analysis, all samples were sputter coated with 10 nm of gold (Au) to improve sample conductivity. Analysis was performed using a ZEISS GeminiSEM 500 field emission scanning electron microscope (FE-SEM). Images were collected with the secondary electron detector at 5 kV to reduce beam/sample interaction volume in accordance with Sakulich et al. [108]. A working distance of 5 mm was selected to increase resolution at high magnifications (8000×). SEM analysis allowed for the characterization of meso- to micro-pore size analysis.

Determination of pore size was conducted using ImageJ software (version 1.54d). Pores within each image were measured twice along the longest diameter and averaged to produce a pore diameter. A minimum of 400 pores were counted for each sample to ensure sufficient sample collection and binning for histogram analysis.

#### 2.3.4. Energy Dispersive Spectroscopy (EDS)

The ZEISS GeminiSEM 500 used for SEM was equipped with an Oxford Instruments X-Max Extreme EDS detector and AZtecLive software (version 6.1). EDS atomic point mapping was performed in tandem with SEM image collection. Therefore, sample preparation and experimental parameters are the same as for SEM. The 10 nm Au sputter coating was selected to avoid overlap with the atomic X-ray emission lines of interest. An accelerating voltage of 5 kV provided sufficient energy (approximately double the largest Kα value of interest) to generate a signal for elemental analysis. All samples were mapped based on the following elements of interest: Al, P, Si, O, S, C, and N. Phosphate geopolymers are known to contain Al, P, Si, and O. The elements C, S, and N were selected in an attempt to observe PBDT and PBDI within the geopolymer matrix. Data were normalized to account for atomic percent incorporation of Al, P, Si, and O.

#### 2.3.5. Scanning/Transmission Electron Microscopy (S/TEM)

Samples were crushed to particle sizes of ≤10 μm and stored under nitrogen prior to analysis. To promote particle separation, samples were suspended in ethanol and deposited onto Ted Pella 200-mesh copper Type B grids. S/TEM analysis was performed using a Thermo ScientificTM Talos F200X scanning/transmission electron microscope. The combination of scanning and transmission electron microscopy allowed for pore size analysis to be performed at the nano-to-meso scale. The high-angle annular dark-field scanning transmission microscope (HAADF-STEM) was coupled with EDS signal detection to generate atomic composition mapping of the geopolymer composite particles. Image collection was taken at an accelerating voltage of 200 kV.

#### 2.3.6. Nitrogen Adsorption (Porosimetry)

Nitrogen (N_2_) adsorption experiments utilized a Micromeritics ASAP 2020 Plus high-performance adsorption analyzer to measure the surface area, pore size, and pore volume of each geopolymer composite. Sample preparation required 2.2–2.5 g of material to be carefully broken into ~3 mm^3^ pieces. The material was stored under a house vacuum at room temperature for 48 h prior to analysis. Additional sample evacuation was performed at 373 K for 240 min under 100 mmHg with an N_2_ atmosphere. Data collection at 77 K and recorded manometrically up to 750 mmHg provided N_2_ adsorption/desorption isothermal curves, which allowed for the application of the Brunauer–Emmett–Teller (BET) method and Barrett–Joyner–Halenda (BJH) theory to calculate surface area and pore size distribution, respectively.

#### 2.3.7. X-ray Diffraction (XRD)

Measurements for XRD were conducted on a Bruker D2 Phaser X-ray powder diffractometer. Data were collected from both bulk and powder samples. Bulk surface samples were polished as described in Section 2.3.3. and mounted on a glass slide. Powder samples were manually ground into particle sizes of ≤10 μm and stored under nitrogen prior to analysis. Approximately 30 mg of each powder sample was placed in a 10 mm × 0.2 mm Rigaku zero-background well holder. Measurements were taken using Cu-Kα radiation (λ = 1.54184 Å, 30 kV, 10 mA). The data range of 2θ was collected from 10° to 90° with a step size of 0.02° and dwell time of 1.0 s per step. Data were baseline corrected with Igor Pro 9 software (version 9.05).

#### 2.3.8. Attenuated Total Reflectance Fourier Transform Infrared Radiation Analysis (ATR-FTIR)

The FTIR spectra were collected using a Perkin Elmer Frontier spectrometer equipped with an ATR-FTIR zinc selenide crystal. Data were collected under ambient conditions from 600–4000 cm^−1^ using 16 scans with a resolution of 1 cm^−1^. All spectra were background subtracted, baseline-corrected, and normalized using the strongest peak at 1101 cm^−1^. All samples were ground to a fine powder of ≤10 μm particle size. Background spectra were collected under ambient conditions with the blank zinc selenide crystal.

#### 2.3.9. Nanoindentation

A digital optical microscope (Mitutoyo 20× objective) was utilized to identify points of interest located on the material. In an effort to characterize the phase-separated inorganic domains, the optical microscope facilitated the identification of Si-rich particulates and Al/P-rich binding domains.

Micromechanical analysis was performed using a Bruker Hysitron TI980 triboindenter fitted with a Berkovitch 50 nm tip. Experiments utilized continuous stiffness mode with a predetermined load of 1 mN and target drift rate of <0.05 nm/s. A fused quartz standard was used before testing to calibrate the instrument and ensure accuracy. All samples were polished to a tolerance of ~1 μm surface roughness. Modulus and hardness values were automatically determined from the unloading curve using methods described elsewhere [108,109]. Each experiment collected 10–12 data points in a 20 μm^2^ area within each elemental-rich domain for all geopolymer composites. This was performed in triplicate. The variation in data points stems from voids, surface roughness, or abnormal pore sizes. Extreme statistical outliers were discarded from the data set.

## 3. Results and Discussion

### 3.1. Solution Properties of PGP Sol-Gel Resins

The water-soluble, high-performance polymers that were used to form PGP hybrid resins are shown in Figure 2. PBDT (linear, *para*-linkage) and PBDI (kinked, *meta*-linkage) are chemically identical regioisomers. Thus, both of the sulfonated, all-aromatic polyamides dissolve in aqueous solutions and maintain stability under acidic conditions. Results from Appendix A demonstrate the compatibility of both PBDT and PBDI in 9M phosphoric acid solutions. At 1.0 wt. %, PBDT in 9M phosphoric acid was a gel at room temperature, while PBDI in 9M phosphoric acid did not demonstrate gel-like behavior until reaching >2.5 wt. %. Due to the significant differences in viscosity between the hybrid resins, the following range of PBDT and PBDI polymer loadings were selected: 0.1, 0.25, and 0.5 wt. % for PGP + PBDT and 0.5, 1.0, and 2.5 wt. % for PGP + PBDI. A control PGP sample was also studied without the addition of a polymer.

Similar to cement pastes, geopolymer resins undergo three major types of physical interactions: electrostatic and van der Waals forces due to colloidal interactions, viscosity contributions as interstitial liquid flows between particles, and friction from direct contact forces [29]. The cumulative effects of these internal forces are provided in Figure 4 in terms of apparent viscosity for PGP + PBDT and PGP + PBDI resins as a function of shear rate. Ranging from low (10^−3^ s^−1^) to intermediate (10^2^ s^−1^) shear rates, the viscosity of all PGP + PBDT and PGP + PBDI resins decreases by greater than four orders of magnitude. Thixotropy (shear thinning) dominates the overall solution behavior. Thixotropy is defined by a time-dependent decrease in viscosity, which is reversible upon the removal of stress or shear [110]. While the characterization of thixotropic fluids using the Cross or Carreau models is common, the inability to measure a zero-shear viscosity at low shear rates only allowed for simple power law fitting. The power law equation is defined as $\eta =k{\dot{\gamma }}^{n-1}$, where *η* is viscosity, k is consistency, $\dot{\gamma }$ is the shear rate, and n is the power law index. Within the measured time range of 10^−3^–10^2^ s^−1^, a power law index of 0.14 + 0.03 was present for all resins. Extremely shear-thinning fluids demonstrate a power law index close to 0, while Newtonian fluids equal 1 because of their independent relationship with respect to time.

Changes in viscosity relative to polymer incorporation appear most prominent within the 0.5 wt. % PBDT resin. The addition of 0.5 wt. % PBDT resulted in a greater than 3× increase in viscosity relative to the virgin PGP resin across all measured shear rates. This increase in viscosity for 0.5 wt. % PBDT also exists relative to all PGP + PBDI resins (Figure 4). We contribute this viscosity relationship to the total number of rigid polyelectrolyte chains in solution. While semi-flexible systems (e.g., sulfonated polystyrene and PBDI) require larger concentration loadings of polyelectrolyte to observe significant changes in viscosity, polymers that form rigid assemblies in solution (e.g., Xanthan and PBDT) reveal entanglement-like behavior at extremely low concentrations [111,112,113].

Importantly, an inflection point appears within both systems at 0.1 s^−1^ (Figure 4). It is understood that the rheological properties of structured fluids at low strain rates are dominated by van der Waals forces, while at higher strain rates, hydrodynamic and inertial forces begin to emerge [29]. The changes in slope at 0.1 s^−1^ likely arise from the electrostatic contribution of the sodium counter ions and polymer chains. Therefore, the addition of 0.25 wt. % polyelectrolyte and above appears to reduce the electrostatic interactions at low shear rates. This behavior stems from the screening effect of sodium ions located within the polymers [114,115]. However, at increased shear rates, the large polymer chains contribute hydrodynamically through overlap and jamming interactions that slightly increase the solution viscosity relative to the virgin PGP system [116,117]. However, the inflection point appears to shift to higher shear rates (10 s^−1^) in the control PGP and 0.1 wt. % PGP + PBDT samples. We attribute this behavior to the low density of interacting polymer chains, leading to lower viscosity measurements. The slight decrease in viscosity within the 0.1 wt. % PGP+PBDT sample relative to the control PGP resin likely stems from the surfactant nature of sulfonated molecules [118].

Prior to oscillatory analysis, the linear viscoelastic regime (LVE) was confirmed with amplitude sweep studies. Storage (G′) and loss (G″) modulus values for both PGP + PBDT and PGP + PBDI resins in Appendix A reveal LVE plateaus between 0.005–0.05% for all samples. Thus, all oscillatory measurements utilized a 0.05% strain oscillation to maintain the solution microstructure of each PGP + Polymer resin during testing.

The dynamic moduli of PGP + Polymer resins were determined via oscillatory frequency studies to probe the viscoelastic behavior as a function of polymer incorporation. Compared to the PGP control resin in Figure 5, the PGP + PBDT 0.1 wt. % hybrid resin revealed a 4–8× reduction in storage modulus across the measured angular frequency (ω) range of 0.1–500 rad·s^−1^. Due to the limited angular frequency range, it is difficult to characterize the terminal behavior of the control and PGP + Polymer resins at low frequencies (below the crossover frequency). The selected range demonstrated that the 0.1 wt. % PBDT resin resulted in a ~10× decrease in modulus with a less than ½ decade reduction in crossover frequency. Importantly, the selected PBDT loadings of 0.1–0.5 wt. % were similar to that of commercial superplasticizers used in concrete mixtures for Portland cement (usually 0.5–1.0 wt. %). Many superplasticizers rely on the electrostatic repulsion of cement particles to increase the zeta potential, resulting in improved workability with up to a 30% reduction in water demand [118].

The 0.25 and 0.5 wt. % PBDT hybrid resins exhibited a systematic increase in G′ values at angular frequencies >10 rad·s^−1^ and maintained G′ over G″ across the entire measured frequency range. The extended period in which G′ exists above G″ suggests the formation of a weak, physically cross-linked gel within the hybrid resins (Figure 5). The crossover frequency of each hybrid resin is predicted to emerge below 0.1 rad·s^−1^, which is more than an order of magnitude below the crossover for the virgin PGP resin. Additionally, the inverse relationship between crossover frequency and relaxation time (τ) helps to illustrate the large change in the processability of the 0.5 wt. % PBDT hybrid resin. As the 0.5 wt. % PBDT system displays both an increase in τ and a significant increase in modulus, the current processing methods are insufficient to target larger loadings of PBDT.

The dynamic moduli of the PGP + PBDI hybrid resins in Figure 5 reveal less significant changes in the viscoelastic properties. All resins (0.5, 1.0, and 2.5 wt. % PBDI) display rubbery/gel-like behavior with G′ over G″ from 0.1–500 rad·s^−1^. A slight increase in modulus is observed as a function of polymer incorporation; however, the deviation spans less than ½ decade in magnitude at 1 rad·s^−1^. Similar to the PGP + PBDT system, all PGP + PBDI samples reveal improved modulus relative to the virgin PGP resin below 1 rad·s^−1^. The crossover frequency can be estimated to occur below 0.1 rad·s^−1^. Considering the rubbery plateau is observed for all resins containing 0.25 wt. % polymer and above, we attribute the increase in modulus relative to the virgin PGP system to the increase in polymer mass (concentration) within the resin.

The relationship between G′ and G″ is further demonstrated in Figure 6, with complex viscosity ($\eta^{*}$) plotted as a function of angular frequency. Complex viscosity is derived from the complex modulus ($G^{*}$) as:(1)$$\left|\eta^{*}\right|={\left({\eta^{\prime }}^{2}+{\eta ”}^{2}\right)}^{\frac{1}{2}}={\left[{\left(\frac{G”}{\omega }\right)}^{2}+{\left(\frac{G^{\prime }}{\omega }\right)}^{2}\right]}^{\frac{1}{2}}=\frac{1}{\omega }\left|G^{*}\right|$$
where $\eta^{\prime }$ is the dynamic viscosity and $\eta^{\prime \prime }$ is the storage viscosity [119]. While continuing to maintain the microstructure of the PGP + Polymer solution, $\eta^{*}$ reveals the enhanced tunability of the hybrid resins from PBDT incorporation. The PBDI loadings result in a less significant contribution to the complex viscosity. The complex viscosity of the PGP + PBDT hybrid resins varies by greater than 10× between the 0.1 to 0.5 wt. % polymer loadings. It is understood that hydrated particles, as a function of electrostatic, hydrodynamic, and hard–hard interactions, play a vital role in the rheological properties of sol-gel resins [28]. Thus, having maintained a constant water-to-solids ratio of inorganic materials, low polymer loadings (0.1 wt. % PBDT) are shown to act as surfactants to reduce particle–particle interactions. However, at sufficient polymer loadings (>0.5 wt. %), increases in $\eta^{*}$ are observed. Large polymer loadings result in slower chain dynamics due to polymer overlap and resin density. Although significant changes in η were not observed for hybrid resins other than the 0.5 wt. % PBDT system, the plotting of $\eta^{*}$ illustrates the improved tunability of the resins with polymer loadings at 0.1 and 0.25 wt. % PBDT (Figure 6). Thus, a great deal of attention must be utilized while designing a high-performance PGP + Polymer composite due to the variability in the processing and handleability of the sol-gel resin.

### 3.2. Composition of PGP Composites

Phosphate geopolymer systems are known to remain thermally stable at temperatures above 1000 °C [36]. The thermal stability of these glass-ceramic materials is achieved through the formation of a 3D inorganic polymer network that results from a low temperature cure at 80 °C and subsequent heat treatment at 260 °C (Figure 3). In the case of PGP + Polymer composites, the organic polymer is physically incorporated into the PGP as a class I geopolymer composite (no strong bonding between organic and inorganic components). Unlike the current state of most class I geopolymer composites, PBDT and PBDI are mixed homogenously throughout the solution and remain physically incorporated in the cured geopolymer plaque [6,68,79].

The overall composition of the PGP + Polymer plaques was confirmed using thermal gravimetric analysis (TGA). Results in Table 3 reveal the thermal properties of PBDT and PBDI, along with the final compositions of the PGP + Polymer plaques. The 5% mass loss (T_d_, 5%) from the thermal decomposition of PBDT and PBDI was 489 °C and 437 °C, respectively. Both polymers are hygroscopic with a total water uptake between 16 to 18% and produce char yields of 27.1 wt. %. To calculate the mass of polymer in the PGP + Polymer system, the mass at 975 °C in the air of each composite was subtracted from the mass of the virgin PGP plaque under the same conditions. Of note, all plaques exhibited a 2 to 3.5 wt. % loss in mass up to 975 °C. This mass loss begins at 300 °C, suggesting the final firing generated additional condensation reactions of the geopolymer, resulting in full conversion and densification. The remaining mass loss is attributed to the degradation of PBDT and PBDI. The temperature ramps and thermal curves of the polymers and PGP + Polymer plaques are provided in Appendix A.

The crystallographic structures of the PGP and PGP + Polymer plaques were characterized using powder X-ray diffraction (XRD). Diffraction patterns of the aluminosilicate and phosphate starting materials are displayed in Appendix A. The patterns track the crystallographic structure of the original kaolinite starting material to the final geopolymer structure after curing and heat treatment to 260 °C. The original kaolinite material (PowerPozz^TM^) transitions from crystalline to disordered after calcination at 700 °C for 2 h. It is understood that calcined kaolinite (metakaolin) appears as semi-crystalline beginning at 500 °C, and transitions to amorphous at temperatures of 550 °C and above [120]. Natural kaolinite contains a ~2% impurity of TiO_2_, resulting in crystalline anatase and rutile diffraction peaks [106,121,122]. After processing and a final heat treatment at 260 °C, the PGP plaque undergoes a crystalline phase transition. As the starting SiO_2_/Al_2_O_3_ material was amorphous, the crystallinity arises from the formation of AlPO_4_ lattice structures. The transition into a semi-crystalline material helps to classify this geopolymer composite as a glass-ceramic, while the low-temperature, ambient processing is unique to cement materials.

The three distinct diffraction peaks in Appendix A within the PGP plaque appear at the 2θ values of 20.50°, 21.63°, and 23.10°. An additional peak at 35.73° is also present. The lattice structure is formed from the polycondensation of Al and P species into AlPO_4_ Tridymite. Bragg reflections of Tridymite have been reported previously: 200 (20.60°), 002 (21.70°), 201 (23.33°), and 212 (35.28°) [123]. Importantly, the crystal structure and Tridymite formation of the virgin PGP plaque remains unchanged in the PGP + Polymer composites (Appendix A) [121,122,124,125,126]. This lack of structural change likely stems from (1) insufficient polymer loading to impact the lattice structures during geopolymerization and (2) the polymer acts as a class I reinforcement agent, meaning the polymer did not contribute to the covalent network of the inorganic materials.

The chemical structure was further characterized using FTIR analysis. The broad peaks at 3700–3100 and 1636 cm^−1^ in Appendix A account for trace amounts of physisorbed water located in the nano-pores of the material, which are slightly more pronounced in the PGP and PGP + Polymer samples. The strong and broad peak at 1250–950 cm^−1^ in all samples originates from a range of aluminosilicate bonds. The broadness and intensity of this peak will vary depending on the percent contribution of each Al-O-P, Al-O-Si, and Si-O-Si stretch to the overall material [127]. The additional shoulder located at 938 cm^−1^ in the geopolymer plaques is a contribution from the Si-O-P bend as the Si-O particles of the starting metakaolin material react with the P-O phosphoric acid tetrahedral units [128]. The suppression of the Si-O-Al metakaolin peak at 800 cm^−1^ supports the biphasic nature of the final PGP plaques. It is understood that the aluminum species within the starting material dissolve and react more quickly than the silicon species [21]. The suppression in the intensity of the Al-O-Si vibration at 800 cm^−1^ clearly illustrates that process. In addition to the suppression of the Al-O-Si metakaolin peak, a new peak at 713 cm^−1^ emerges. Although sources vary in regard to the exact contribution of this peak, the signal likely stems from a Si-O-X (X = Al, P, Si) bond [36,129]. Importantly, similar to the XRD crystal structure, the chemical structure of the PGP and PGP + Polymer plaques appear unchanged. Again, there is likely insufficient polymer present to alter the chemical structure and the polymer did not covalently bond with any of the inorganic materials. This is supported in Appendix A by the identical FTIR signals in the PGP + Polymer composites relative to the virgin PGP material.

Additional elemental characterization was performed using EDS analysis. It is understood that the phosphoric-acid-catalyzed geopolymer synthesis occurs mainly due to the reaction between phosphorous (phosphoric acid) and aluminum (metakaolin) [26,30]. During the dissolution and mixing process, the PO_4_^+5^ cationic tetrahedral species interact with the anionic AlO_4_^−5^ species to balance charges and maintain neutrality. This process occurs as Al^3+^ leaches from the metakaolin and precipitates as AlPO_4_. Thus, the Al/P-rich domains in Figure 7 are referred to as the binding phase, while the Si-rich domains are referred to as the particulate phase. Additional elements representative of the organic polymer, such as C, N, and S, were selected for analysis; however, the signal-to-noise ratio was insufficient to accurately determine the location of the polymers within the geopolymer composite. Higher weight percent loadings of organic polymer (>2.5 wt. %) may be necessary to detect elements separate from the geopolymer matrix.

While the SEM micrographs in Figure 7 display varying degrees of porosity in the PGP (no polymer), PGP + PBDT 0.5 wt. %, and PGP + PBDI 2.5 wt. % plaques, elemental mapping reveals similar phase domains. The Si Kα X-ray emission energies for all samples overlay with the dense (darker) regions in the SEM micrograph. In contrast, the Al and P Kα X-ray emission energies overlay with the porous (light) regions of the PGP (no polymer) and PGP + PBDI 2.5 wt. % plaques. Identification of porous domains within the PGP + PBDT 0.5 wt. % plaques appears more challenging; however, the elemental maps provide clear indications of phase separation of inorganic domains. The Si-rich particulates and Al/P-rich binding domains in all PGP and PGP + Polymer plaques suggest that the reaction mechanism remains unchanged regardless of viscosity and polymer incorporation.

EDS analysis was also performed using a transmission electron microscope (TEM). TEM micrographs and Si, Al, and P Kα X-ray emission elemental maps in Appendix A demonstrate a similar biphasic structure as demonstrated during SEM analysis.

The atomic percent composition of each material was determined during EDS analysis. Atomic percent values (at. %) of all geopolymer plaques are provided in Table 4 and Appendix A. The targeted atomic % ratio of 66.7(O):11.1(Si, Al, P) was used for all PGP and PGP + Polymer plaques. When averaged across all domains, each element lies within a single standard deviation of the targeted ratio. However, as demonstrated in the SEM/TEM micrographs and EDS elemental maps, Si-rich and Al/P-rich domains are clearly present. The denser particulate phases contain approximately 20 at. % silicon. The remaining 7 at. % of aluminum and phosphorous within the Si-rich particulate further supports the enhanced solubility and dissolution of aluminum relative to silicon. The final molar ratios from TEM/EDS analysis are approximated as follows: Si-rich = 5:2:2 Si:Al:P (5 SiO_2_, 1 Al_2_O_3_, 1 P_2_O_5_); Al/P-rich = 2:5:5 Si:Al:P (4 SiO_2_, 5 Al_2_O_3_, 5 P_2_O_5_).

### 3.3. Porosity of PGP Composites

Pore size analysis was performed across multiple length scales using SEM and S/TEM analysis, as displayed in Table 5. The micrographs in Figure 8 and Figure 9 reveal porous domains within the Al/P binding domains of the virgin PGP plaque. These porous structures develop as aluminum leaches out from the aluminosilicate metakaolin starting material and polymerizes with the aqueous phosphoric acid [40]. Pore size distributions are provided as histograms in Figure 8 and Figure 9, with the averages and medians calculated in Table 5. Median values are displayed due to the non-normal (asymmetric) distribution of pore sizes.

Relative to the virgin PGP plaque, similar porous structures are observed for the PGP + PBDT 0.1 and 0.25 wt. % composites (Figure 8). However, the pore size analysis of the PGP + PBDT 0.5 wt. % plaque displays SEM micrographs with a majority of pore sizes < 0.4 μm. Based on the structural and elemental analysis, the PGP + PBDT 0.5 wt. % plaque revealed no significant chemical difference relative to the virgin PGP plaque. Thus, we hypothesize the reduction in porosity within the PGP + PBDT 0.5 wt. % plaque stems, in part, from the rheological properties. As the 0.1 and 0.25 wt. % PBDT resins reduced $\eta^{*}$ and display little influence on the apparent viscosity, the 0.5 wt. % PBDT resin revealed an overall increase in η, $\eta^{*}$, and modulus. During curing, the large increase in G′ over G″ likely mitigates the strong phase separation of the Al/P binding domain into porous structures, allowing for a well-dispersed system. Additional in situ rheology/curing studies will assist in pursuing this hypothesis.

The pore structures of the PGP + PBDI composites in Figure 9 appear to produce a greater variance in pore sizes. Relative to the PGP + PBDT 0.5 wt. % composite, the PGP + PBDI 0.5 wt. % composite did not produce an overall decrease in pore size. While the rheological properties appear correlated with porosity in the composite plaques, a further investigation of the in situ polymer–geopolymer interactions and observation of the evolution of microstructures during processing will assist in isolating the role of the polymer in the resin and solid state. The changes in pore sizes are shown in Figure 9 and Table 5. An increase in average pore size from 564 to 942 nm was observed for 0.5 and 2.5 wt. % PBDI plaques, respectively. While the viscosity of the PGP + PBDT resins above 0.5 wt. % proved extremely difficult to process, the viscosity of the PBDI system indicates that loadings above 2.5 wt. % may still be feasible to process.

### 3.4. Micromechanical Properties of PGP Composites

Micromechanical characterization was performed via nanoindentation experiments. Analysis of the Al/P-rich binding phases and Si-rich particulate domains utilized a 50 nm Berkovitch indentation tip. Data were collected within a 20 μm^2^ area for each elemental-rich domain. The results in Figure 10 demonstrate similar modulus and hardness values when comparing Si-rich and Al/P-rich domains within the same plaque. The virgin PGP modulus (15.48 ± 3.41 GPa) and hardness (1.00 ± 0.28 GPa) values are similar to other previously reported geopolymer systems [53,108,130]. At heat treatment temperatures between 200–700 °C, Beleña and Zhu recorded modulus and hardness values of 16 GPa and 0.7 GPa, respectively [130]. Importantly, T-test probability values (95% confidence) suggest that the modulus of the Si-rich domains within the PBDI 1.0 and 2.5 wt. % composites is statistically larger than that of the Al/P-rich binding phase. This change in modulus may be caused by (1) the broad micro-porosity within the plaques and (2) the decrease in nano-porosity within the Si-rich domains.

Additional T-test studies (95% confidence) were performed to determine if a significant difference exists between the virgin PGP plaque and the PGP + Polymer composites. Other than the PGP + PBDT 0.5 wt. % composite, all PGP + Polymer composites demonstrate a statistically significant improvement in both modulus and hardness relative to the virgin PGP system (Figure 10). While polymer loadings of 0.1–1.0 wt. % PBDT and PBDI reveal a 26% modulus and 31% hardness increase, polymer loadings of 2.5 wt. % PBDI suggest more significant improvements in micromechanical properties. The 2.5 wt. % PGP + PBDI composite resulted in an average modulus increase of 50% and a hardness increase of 91% across the bulk of the material. In contrast to fiber-reinforced materials, these polymer-modified geopolymers demonstrate an overall improvement in micromechanical properties irrespective of polymer location. Sakulich and Li utilized polyvinyl alcohol (PVA) fibers as a reinforcement material in a fly-ash-based cementitious composite. Modulus values of 15.5 GPa were reported close to fibers (5 μm), and larger modulus values of 43.2 GPa emerged at distances of 50 μm from the fibers [108]. Due to the hydrophilicity of the PVA, the authors suggest the fibers did not solidly bond to the matrix, resulting in incompatibility between the bulk matrix and reinforcement fibers. However, similar reinforcement studies revealed a 4× increase in debonding energy with the addition of 15 wt. % polyacrylate [131]. Polymers are known to act across different length scales and domain sizes, which fill voids and improve interface strengths between phases. Overall, the addition of water-soluble, high-performance polymers revealed no negative effects on the micromechanical properties of the PGP composites. Of future interest will be the casting of PGP + PBDI plaques above 2.5 wt. % and the pursuit of class II geopolymer composites to further improve the compatibility between phase-separated domains.

## 4. Conclusions

Neat PGP and PGP + Polymer sol-gel resins were prepared from metakaolin and 9M phosphoric acid with a water-soluble, high-performance solution. The rheological properties of the sol-gel resins were studied prior to mold casting and hardening at 80 °C with a final 260 °C heat treatment. The PGP composites contained linear PBDT (0.1, 0.25, and 0.5 wt. %) and kinked PBDI (0.5, 1.0, and 2.5 wt. %). Characterization of the hardened plaques utilized TGA, FTIR, XRD, EDS, SEM, TEM, BET, and nanoindentation.

Rheological analysis demonstrated a range of solution behaviors within the polymer concentration ranges of 0.1–2.5 wt. %. Similar to commercial cement superplasticizers (0.5–1.0 wt. %), PGP + PBDT 0.1 wt. % revealed a 4–10× decrease in modulus and complex viscosity relative to the virgin PGP resin. In contrast, the linear PBDT chains at 0.5 wt. % contributed to a 3× increase in apparent viscosity and a 1–10× increase in modulus and complex viscosity. The selected PBDI concentrations also demonstrated a ~3× increase in modulus at low frequency rates; however, a ~2× decrease was observed at higher frequency rates compared to the neat PGP resin. The ranges of polymer loadings were selected based on processability; higher loadings resulted in increased viscosities.

Pore size analysis with SEM, TEM, and BET techniques revealed both micro- and nano-porosity control with the addition of a water-soluble, high-performance polymer. SEM analysis of PGP and PGP + PBDT 0.5 wt. % composites revealed a reduction in pore size from 0.78 μm to 0.31 μm, respectively. TEM and BET analysis exhibited a 23–38% decrease in nano-porosity as a result of polymer infiltration.

Micromechanical experiments resulted in Young’s modulus and hardness values similar to those of previously reported geopolymer systems. The incorporation of water-soluble, high-performance polymers resulted in improvements in the order of ~30–40% in both Young’s modulus and hardness.

XRD, FTIR, and EDS analysis did not detect any differences between the neat PGP plaque and PGP + Polymer composites. The similar structural and chemical properties likely stem from (1) the low concentration loadings of polymer and (2) the utilization of PBDT and PBDI as class I geopolymer reinforcement materials (non-covalent, weak intermolecular interactions).

In summary, this work provides a fundamental understanding of the processing, microstructure, and mechanical behavior of phosphate geopolymers and water-soluble, high-performance polymer-reinforced geopolymer composites. These findings suggest that high-performance polyelectrolytes demonstrate potential as rheological modifiers and structural/mechanical reinforcements. Future work of interest includes: (1) targeting higher water-soluble high-performance polymer loadings and (2) pursuing class II covalent polymer reinforcements to further tailor the microstructure and mechanical properties of the resulting geopolymer composite.

## Figures and Tables

**Figure 1 materials-17-02856-f001:**
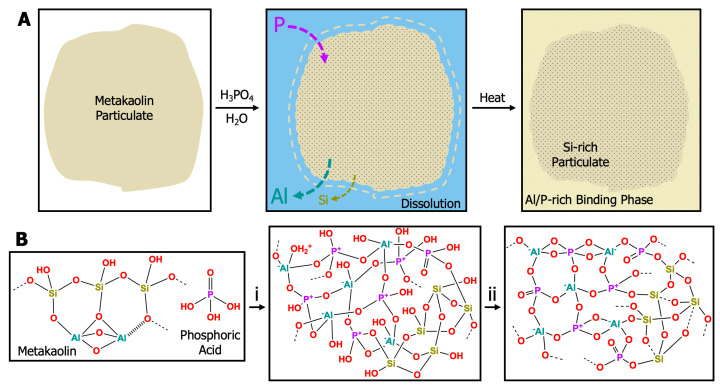
(**A**) Cartoon representation of the dissolution and geopolymerization of a metakaolin particulate with phosphoric acid. (**B**) Schematic representation of the geopolymerization of an acid-catalyzed phosphorous geopolymer into a 3D inorganic network. Metakaolin is mixed with phosphoric acid to create a sol-gel resin. (**B**,**i**) Condensation of PO_4_^3−^, Al^3+^, and SiO species forms oligomers and a hydrated silica network during the cure cycle from room temperature to 80 °C. (**B**,**ii**) Additional heat treatment up to 260–300 °C is required for strength and moisture stability and results in further water loss, evaporation, dehydration, condensation, and crystallization [26].

**Figure 2 materials-17-02856-f002:**
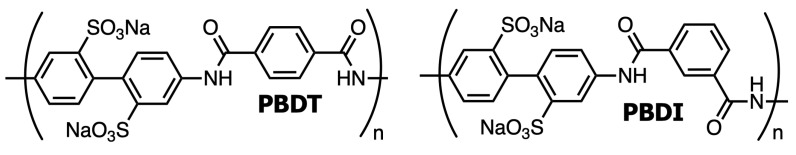
Chemical structures of linear (PBDT) and kinked (PBDI) water-soluble, high-performance polyelectrolytes. PBDT forms a liquid crystal (nematic) phase in water at ~1 wt. % [100].

**Figure 3 materials-17-02856-f003:**
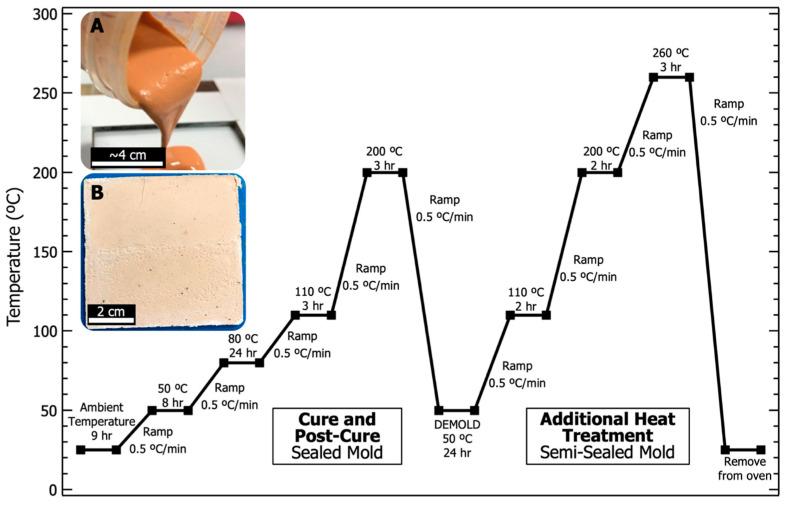
Heating and cooling curves for the curing and heat treatment of all phosphate geopolymer (PGP) plaques. (**A**) Digital photograph of PGP resin with 2.5 wt. % PBDI being poured into the mold. (**B**) Digital photograph of PGP plaque with 2.5 wt. % PBDI after being heat-treated to 260 °C.

**Figure 4 materials-17-02856-f004:**
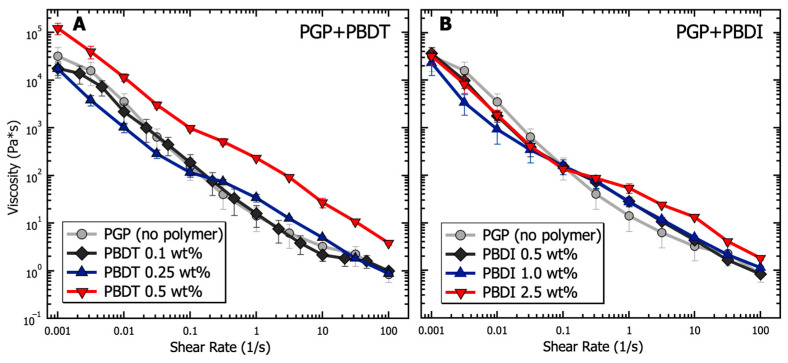
Analysis of the viscosity as a function of shear rate for PGP resins containing (**A**) 0.1, 0.25, and 0.5 wt. % PBDT and (**B**) 0.5, 1.0, and 2.5 wt. % PBDI. Samples utilized 25 mm stainless steel parallel plates, 25 °C temperature control, and all experiments were performed in triplicate.

**Figure 5 materials-17-02856-f005:**
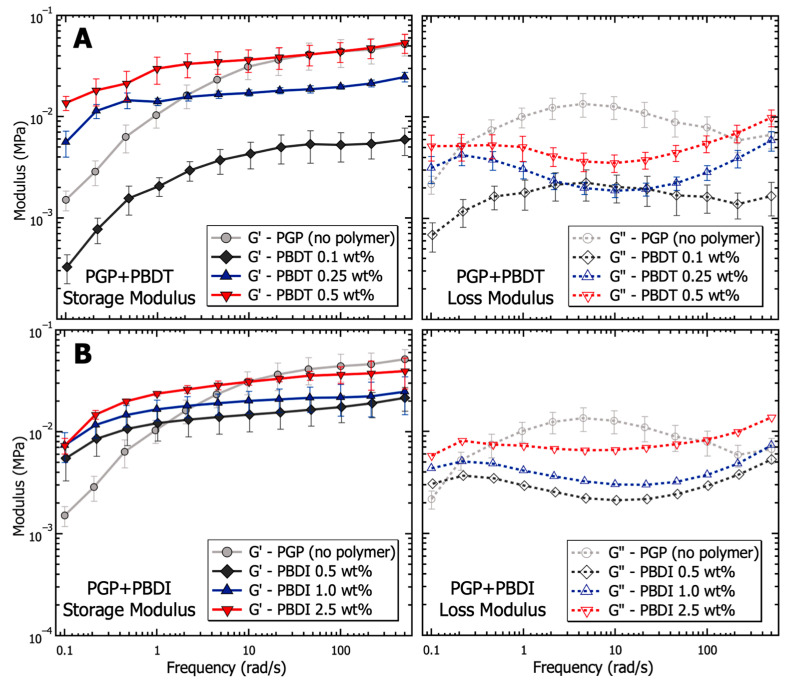
Analysis of the viscoelasticity (storage and loss modulus) as a function of frequency for PGP resins containing (**A**) 0.1, 0.25, and 0.5 wt. % PBDT and (**B**) 0.5, 1.0, and 2.5 wt. % PBDI. Samples utilized 25 mm stainless steel parallel plates, 0.05% strain oscillation, 25 °C temperature control, and all experiments were performed in triplicate.

**Figure 6 materials-17-02856-f006:**
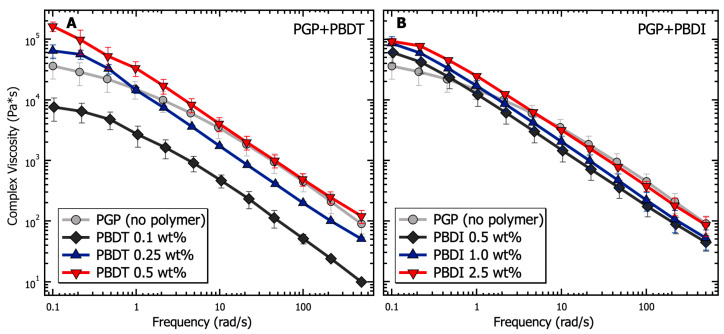
Analysis of the complex viscosity as a function of frequency for PGP resins containing (**A**) 0.1, 0.25, and 0.5 wt. % PBDT and (**B**) 0.5, 1.0, and 2.5 wt. % PBDI. Samples utilized 25 mm stainless steel parallel plates, 0.05% strain oscillation, 25 °C temperature control, and all experiments were performed in triplicate.

**Figure 7 materials-17-02856-f007:**
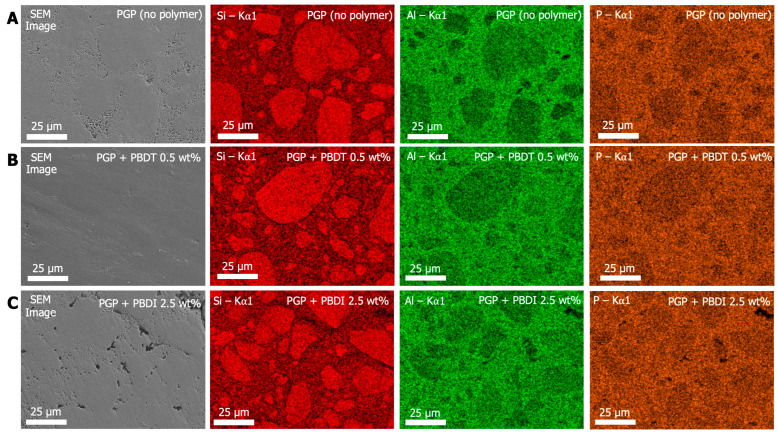
SEM and EDS images of (**A**) PGP (no polymer), (**B**) PGP + PBDT 0.5 wt. %, and (**C**) PGP + PBDI 2.5 wt. %. Colored EDS Kα1 emission maps of Si (red), Al (green), and P (orange).

**Figure 8 materials-17-02856-f008:**
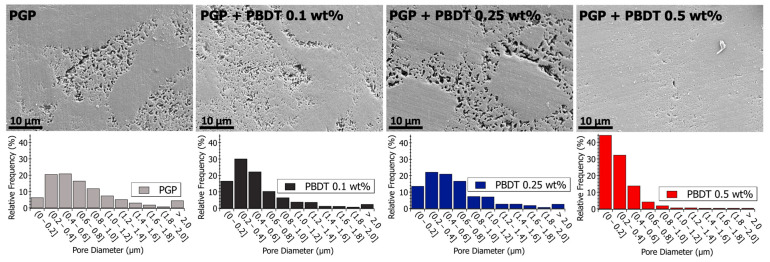
(**Top**) Scanning electron microscopy images of polished PGP and PGP + PBDT plaques. (**Bottom**) Histograms of PGP and PGP + PBDT. Pore size diameter was determined with ImageJ analysis software.

**Figure 9 materials-17-02856-f009:**
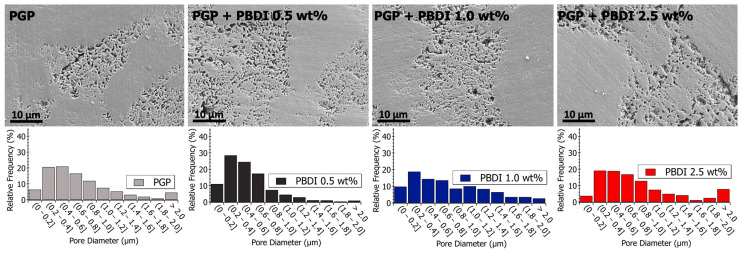
(**Top**) Scanning electron microscopy images of polished PGP and PGP + PBDI plaques. (**Bottom**) Histograms of PGP and PGP + PBDI. Pore size diameter was determined with ImageJ analysis software.

**Figure 10 materials-17-02856-f010:**
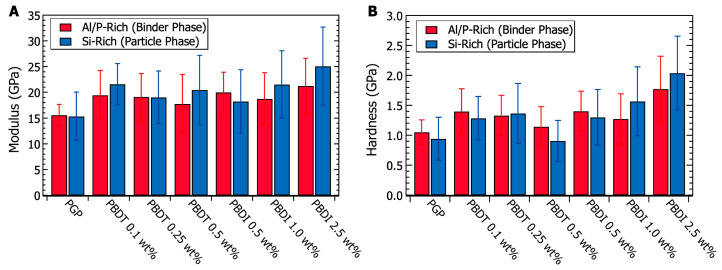
(**A**) Nanoindentation modulus values taken from the binder phase (Al/P-rich domain) and particulate phase (Si-rich domain) of PGP and PGP composites. (**B**) Nanoindentation hardness values taken from the binder phase (Al/P-rich domain) and particulate phase (Si-rich domain) of PGP and PGP composites.

**Table 1 materials-17-02856-t001:** Chemical composition of commercially available PowerPozz™ in weight percent (wt. %).

Atomic Species	SiO_2_	Al_2_O_3_	Fe_2_O_3_	CaO	K_2_O	Na_2_O	P_2_O_5_	TiO_2_	MgO
Composition (wt. %) *	54.04	41.93	1.52	0.13	0.48	0.047	0.15	2.35	0.12

* Data averaged from multiple sources [102,103,104,105,106].

**Table 2 materials-17-02856-t002:** Mixing formulation of phosphate geopolymer (PGP) sol-gel resins.

Sample	Polymer	Weight Percent of Polymer	Mass of Polymer Added	Mass of Metakaolin Added	Mass of 9M Phosphoric Acid Added	Mass of Total Solids
		wt. %	G	g	g	g
PGP	-	-	-	20.000	26.926	31.468
PBDT 0.1	PBDT	0.101	0.027	20.000	26.926	31.495
PBDT 0.25	PBDT	0.254	0.068	20.000	26.926	31.538
PBDT 0.5	PBDT	0.503	0.136	20.000	26.926	31.604
PBDI 0.5	PBDI	0.503	0.136	20.000	26.926	31.604
PBDI 1.0	PBDI	1.003	0.273	20.000	26.926	31.741
PBDI 2.5	PBDI	2.502	0.691	20.000	26.926	32.172

**Table 3 materials-17-02856-t003:** Calculated values of the final weight percentage composition of polymers in plaques.

Sample	T_d, 5%_	Total Water Loss @ 200 °C	Total Mass Loss @ 975 °C	Calculated Mass of Polymer in Composite	Targeted Solids Content of Polymer in Composite
	°C	wt. %	wt. %	wt. %	wt. %
PBDT	489	18.1	72.9	-	-
PBDI	437	16.0	72.9	-	-
PGP	-	4.97	2.04	-	-
PBDT 0.1	-	4.13	2.13	0.092	0.087
PBDT 0.25	-	5.24	2.25	0.216	0.222
PBDT 0.5	-	5.63	2.44	0.397	0.430
PBDI 0.5	-	4.71	2.48	0.445	0.430
PBDI 1.0	-	4.13	2.67	0.632	0.860
PBDI 2.5	-	4.97	3.47	1.435	2.188

**Table 4 materials-17-02856-t004:** Average atomic percent incorporation of all PGP and PGP + Polymer plaques from TEM analysis.

Domains	All Samples				Target Ratio
	O	Si	Al	P	O:Si, O:Al, O:P
	Atomic %	Atomic %	Atomic %	Atomic %	Atomic %
Si-rich	65.37 ± 1.39	19.77 ± 2.71	7.68 ± 2.24	7.17 ± 1.38	66.7:11.1
Al/P-rich	64.97 ± 1.53	5.63 ± 2.05	15.03 ± 0.32	14.38 ± 0.37	66.7:11.1
Average	65.20 ± 1.88	12.42 ± 5.23	11.67 ± 2.79	10.71 ± 2.65	66.7:11.1

**Table 5 materials-17-02856-t005:** Measurements of pore size for PGP, PGP + PBDT, and PBP + PBDI composites.

Sample	SEM	SEM	TEM	TEM	N_2_ Adsorption	N_2_ Adsorption	N_2_ Adsorption
	Average Pore Size	Median Pore Size	Average Pore Size	Median Pore Size	BET Pore Size	BET Surface Area	BJH Pore Volume
	nm	nm	nm	Nm	nm	m^2^/g	cm^3^/g
PGP	784	627	38.1	35.9	23.0	21.4	0.129
PBDT 0.1	570	430	30.0	27.3	18.4	12.7	0.059
PBDT 0.25	658	534	27.3	25.8	12.6	45.2	0.130
PBDT 0.5	313	234	24.5	21.7	15.0	33.2	0.112
PBDI 0.5	564	480	25.6	27.5	10.4	42.8	0.095
PBDI 1.0	820	697	30.6	33.2	12.2	19.8	0.063
PBDI 2.5	942	696	21.0	18.5	15.4	32.2	0.116

## Data Availability

The original contributions presented in this study are included in the article/Appendix A, further inquiries can be directed to the corresponding author.

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
