# Peer review of "Effect of Water-Soluble Polymers on the Rheology and Microstructure of Polymer-Modified Geopolymer Glass-Ceramics"

_materials, 2024, doi:10.3390/ma17122856_

Round 1
Reviewer 1 Report
Comments and Suggestions for Authors
Review manuscript ID materials-3024306: Effect of water-soluble polymers on the rheology and microstructure of polymer-modified geopolymer glass-ceramics.
This paper presents the preparation and the rheological study of geopolymer glass-ceramic composites. A very extensive characterization is carried out through TGA, FTIR, XRD, EDS, SEM, TEM, BET, and nanoindentation techniques. The manuscript is clear, explaining in detail the experimental data and with enough references. This work is very interesting for understanding the processing, microstructure, and mechanical properties of water-soluble polyelectrolyte-reinforced geopolymer glass-ceramics.
It is pleased for me to recommend this work, in the present form, to be accepted for publication in "Materials".
Author Response
Review manuscript ID materials-3024306: Effect of water-soluble polymers on the rheology and microstructure of polymer-modified geopolymer glass-ceramics.
This paper presents the preparation and the rheological study of geopolymer glass-ceramic composites. A very extensive characterization is carried out through TGA, FTIR, XRD, EDS, SEM, TEM, BET, and nanoindentation techniques. The manuscript is clear, explaining in detail the experimental data and with enough references. This work is very interesting for understanding the processing, microstructure, and mechanical properties of water-soluble polyelectrolyte-reinforced geopolymer glass-ceramics.
It is pleased for me to recommend this work, in the present form, to be accepted for publication in "Materials".
The authors appreciate your time and consideration. Thank you for your comments and feedback.
Reviewer 2 Report
Comments and Suggestions for Authors
Dear Authors,
Thanks for studying the contribution of geopolymers to glass-ceramic composites. Manuscript is interesting and well organized. However, I have these comments that may enhance its readability:
1) Description of the polymers lacks a graphical and real representation in the main text, although there are insets in Figure S1. It is suggested to move Figure S1 to the main text to improve its understandability.
2) It would be interesting to analyze the correlation between porosity and flexural or bending modulus, since nanoindentation might provide too local information in this particular case.
3) Further explanation about why reveling average and median values in Table 5 (may be related to the histograms' asymmetry) is recommended.
Best regards.
Comments on the Quality of English LanguageI have found these typos:
- Lines 21, 172, 238, 250, 279, 305, 317, 329, 331, 584, 629, 653, 654, and 687: "µm".
- Line 418 and 609: extra space.
- Line 493: "2" in "TiO2" as a subscript.
Author Response
Dear Authors,
Thanks for studying the contribution of geopolymers to glass-ceramic composites. Manuscript is interesting and well organized. However, I have these comments that may enhance its readability:
1) Description of the polymers lacks a graphical and real representation in the main text, although there are insets in Figure S1. It is suggested to move Figure S1 to the main text to improve its understandability.
Figure S1 has been moved to main text and is labeled as Figure 3 (lines 220-224). All figures following have been renumbered in the main text and supporting information.
2) It would be interesting to analyze the correlation between porosity and flexural or bending modulus, since nanoindentation might provide too local information in this particular case.
The main aim of this study was to probe the fundamental class-I geopolymer composite relationships. Specifically, the relationship between inorganic geopolymer materials and a novel class of polyelectrolyte additives. While we agree that the impact on flexural or bending properties is of interest, the collection of bulk mechanical properties falls outside of the scope of this report. This would involve synthesis of larger quantities of polyelectrolyte material to generate a sufficient amount of data points to collect bulk material properties. However, this was not possible under current funding and time restrictions. Follow-on efforts are planned which may include bulk mechanical characterization of polymer-geopolymer materials.
3) Further explanation about why reveling average and median values in Table 5 (may be related to the histograms' asymmetry) is recommended.
Explanation has been provided in the main text (lines 586-588). Due to the non-normal (asymmetric) distribution of pore sizes in figures 8 and 9, we provide both average and median values in table 5.
Best regards.
Comments on the Quality of English Language
I have found these typos:
- Lines 21, 172, 238, 250, 279, 305, 317, 329, 331, 584, 629, 653, 654, and 687: "µm".
All "micrometer" typos have been corrected in the PDF version of this manuscript.
- Line 418 and 609: extra space.
All ‘extra space’ typos have been corrected in the PDF version of this manuscript.
- Line 493: "2" in "TiO2" as a subscript.
The “2” in “TiO2” has been corrected to TiO2 in line 499.
The authors appreciate your time and consideration. Thank you for your comments and feedback.
Reviewer 3 Report
Comments and Suggestions for Authors
Dear Authors,
Manuscript dealing with phosphate geopolymer glass-ceramic composites provides a thorough experimental overview. Proper introduction and clear experimental description support the readability. Still, some cosmetic corrections are needed:
1. In several places (rows 21,172, 238, 250, 279, 305, 317, 329, 331, 630, 653, 654) micrometer character is missing.
2. Figures 7 and 8 histograms should contain error bars.
Those are minor corrections to improve readability.
Author Response
Dear Authors,
Manuscript dealing with phosphate geopolymer glass-ceramic composites provides a thorough experimental overview. Proper introduction and clear experimental description support the readability. Still, some cosmetic corrections are needed:
- In several places (rows 21,172, 238, 250, 279, 305, 317, 329, 331, 630, 653, 654) micrometer character is missing.
All "micrometer" typos have been corrected in the PDF version of this manuscript.
- Figures 7 and 8 histograms should contain error bars.
Adding error bars to a histogram is not necessary as relative frequency calculates . Where is the number of events (pores in a size range) and is total number of events (total number of pores). Error bars cannot be added to the histogram columns, the histogram itself is providing a visualization of the distribution.
Those are minor corrections to improve readability.
The authors appreciate your time and consideration. Thank you for your comments and feedback.
Reviewer 4 Report
Comments and Suggestions for Authors
The paper studies the effects of polymer reinforcement for phosphate geopolymer glass- ceramic composites, mainly focusing on the modifications occurring to the rheological and microstructure properties.
Despite the large number of techniques used to study the prepared samples, some differences are observed only by studying the rheology and the porosity.
For other techniques the lack of differences with the starting material is attributed to the low amount of used additives, so the provided information seem to be less meaningful. This point should be discussed.
Therefore, the obtained results could be interesting but are not exhaustive in the present form of the manuscript. I suggest to modify the work according to the following observations
- At lines 342-345 authors explain that the used polymer loadings were selected according to differences in the viscosity between the hybrid resins. This point should be clarified and further discussed. Indeed the use of a larger amount of loadings could have been useful to understand the mechanisms of interactions between the two components.
- Figure 3: the sample with a 0.1wt% of PBDT does not show a change of slope at 0.1 s-1, which indeed seems shifted at higher values, 10s-1.Moreover the same sample presents peculiar behavior also in Figure 4 panel a and in Figure 5 panel a. This points are should be discussed.
- -Lines 475-479: I don’t get why the authors suggests the occurrence of condensation reactions. This point should be clarified.
- Table 4: why do they show average values? I suggest to show data for each samples, even if in the SI.
- Lines 593-600: I suggest that the measured rheological properties are the effect of the different porosity, not the opposite,
- Fig S3, panel A, the caption has a typo (PBDI instead of PBDT)
- Fig. S5, panel a. I suggest to change the scale, showing data up to 1500cm-1 to enlarge the observed absorption and to make the comparison easier.
- Fig S6, the caption should be revised
A general discussion on how the polymer modify the properties of the composite material should be added, using presently reported results.
Comments on the Quality of English Language
Moderate editing of English language is required
Author Response
The paper studies the effects of polymer reinforcement for phosphate geopolymer glass- ceramic composites, mainly focusing on the modifications occurring to the rheological and microstructure properties.
Despite the large number of techniques used to study the prepared samples, some differences are observed only by studying the rheology and the porosity.
For other techniques the lack of differences with the starting material is attributed to the low amount of used additives, so the provided information seem to be less meaningful. This point should be discussed.
The limited impact of low polymer additives is discussed in the Introduction (lines 159-162) and Conclusion (lines 697-701). We acknowledge that low amounts of polymer additives resulted in the inability to detect changes in XRD, FTIR, and EDS. However higher loadings result in viscosities that are more difficult to process. Significant changes to crystal and chemical structures are also not expected in class I hybrid materials due to a lack of strong organic-inorganic bonding interactions. The scope of this report focuses on class I hybrids, and subsequent work on polymer-geopolymer class II hybrids is planned.
Therefore, the obtained results could be interesting but are not exhaustive in the present form of the manuscript. I suggest to modify the work according to the following observations
- At lines 342-345 authors explain that the used polymer loadings were selected according to differences in the viscosity between the hybrid resins. This point should be clarified and further discussed. Indeed the use of a larger amount of loadings could have been useful to understand the mechanisms of interactions between the two components.
PBDT gelled at 1.0wt%; PBDI behaved gel-like above 2.5wt%. Further explanation of the differences in viscosity is provided in lines 345-348 in the main text.
- Figure 3: the sample with a 0.1wt% of PBDT does not show a change of slope at 0.1 s-1, which indeed seems shifted at higher values, 10s-1.Moreover the same sample presents peculiar behavior also in Figure 4 panel a and in Figure 5 panel a. This points are should be discussed.
Further explanation provided in lines 384-386 for figure 4 (former figure 3).
0.1 wt% PBDT is discussed in figure 5 (former figure 4) in lines 399-409.
0.1 wt% PBDT is discussed in figure 6 (former figure 5) in lines 451-456.
- -Lines 475-479: I don’t get why the authors suggests the occurrence of condensation reactions. This point should be clarified.
It is known in literature that aluminum phosphate-based materials do not fully dehydrate/ condense until ~200-260 °C. At lower temperatures hydrated phases such as variscite exist. Aluminum solubility also increases as a function of temperature but is of course dependent on water content.
See ref. 26 and ref. 15 for example: Morris, J. H.; Perkins, P. G.; Rose, A. E. A.; Smith, W. E., The chemistry and binding properties of aluminium phosphates. Chemical Society Reviews 1977, 6 (2), 173-194., Gilliard Hensel, N.; Franz, G.; Riedl, M.; Gottschalk, M.; Wunder, B.; Galbert, F.; Nissen, J. Polymorphism and Solid Solution in the System SiO2-AlPO4(-H2O): A Review and New Synthesis Experiments up to 3.5 GPa and 1573 K. Neues Jahrb. fur Mineral. Abhandlungen 2007, 184, 131–149, doi:10.1127/0077-7757/2007/0087. See also Wagh, A. S.; Chemically Bonded Phosphate Ceramics 2nd ed, CHAPTER 11, Aluminum Phosphate Ceramics, 2016, ISBN: 9780081003800
In this manuscript and supporting information we demonstrate how water is removed up to 200 ºC, additional mass loss begins at 300 ºC (polycondensation), and a majority of mass loss above 400 ºC due to polymer, though some small amount of residual geopolymer condensation may be occurring in this range as well (in some cases the condensation of OH groups to metal oxide bonds is reported at temperatures above 400 °C). Additional explanation provided in lines 482-485.
- Table 4: why do they show average values? I suggest to show data for each samples, even if in the SI.
EDS chemical composition has been provided in Table S2.
- Lines 593-600: I suggest that the measured rheological properties are the effect of the different porosity, not the opposite,
Rheology was conducted on the viscous resins. The resins set at low temperatures (RT-80°C) to an amorphous solid lacking significant micron-scale porosity. Above ~110C crystallization occurs and micron-scale porosity forms. This porosity is not present in the resin state but evolves in the solid state as a function of temperature (we are unsure of the nano-scale porosity present in the resin state). While the rheological properties are correlated with heat treated porosity content, it is difficult to say without further evidence whether the porosity content and distribution is strictly a rheological effect.
Refs on phosphate geopolymer behavior:
- William Monzel, Olivia Meyer, Kyle Schroeder, Allison Hohenshil, Adam Rape, Kathryn Doyle, Devon M. Samuel, Waltraud M. Kriven “Investigations of Phosphate Geopolymers” Composites and Materials Expo, Anaheim Ca, Oct. 19, 2022 https://digitallibrarynasampe.org/data/webpages/c2022_webpages/TP22-0000000145.html
- Jacob Monzel, Waltraud M. Kriven, Greeshma Gadikota, Gregory Neher, Devon Samuel, Allison Hohenshil, Hassnain Asgar “Investigations of SilicoAluminoPhosphate Geopolymer Glass-Ceramics” International Conference & Exposition on Advanced Ceramics & Composites, Jan 27, 2022
Revisions to the main text have been provided in lines 606-611.
- Fig S3, panel A, the caption has a typo (PBDI instead of PBDT)
The typo has been revised to state “Water uptake of PBDT and PBDI”.
- Fig. S5, panel a. I suggest to change the scale, showing data up to 1500cm-1 to enlarge the observed absorption and to make the comparison easier.
The purpose of figure S4, panel a (formally Fig. S5, panel a) is to demonstrate the full FTIR sprecta collected for each geopolymer composite sample. As discussed in the main text (lines 515-518), the weak and broad peaks within the 4000-1500 cm-1 range are almost insignificance and result from physisorbed water.
- Fig S6, the caption should be revised
The caption in Fig. S5 (formally Fig. S6) has been revised to more clearly communicate the colors, scales, and sections.
A general discussion on how the polymer modify the properties of the composite material should be added, using presently reported results.
A brief discussion regarding the significance of polymer-modified geopolymers has been provided in the conclusion. Lines 702-709 detail the role of polymer modification in geopolymer composites. Additionally, lines 704-706 have been revised to clearly state the potential of these materials based on findings from this report.
The authors appreciate your time and consideration. Thank you for your comments and feedback.
Round 2
Reviewer 4 Report
Comments and Suggestions for Authors
The authors addressed a lot of may comments. I am not fully persuaded of a couple of their answers, so I ask them to further improve the manuscript according to the following points:
- The authors reply to my concerns about the poor modifications induced by a low amount of polymer additives (We acknowledge that low amounts of polymer additives resulted in the inability to detect changes in XRD, FTIR, and EDS. However higher loadings result in viscosities that are more difficult to process. Significant changes to crystal and chemical structures are also not expected in class I hybrid materials due to a lack of strong organic-inorganic bonding interactions. The scope of this report focuses on class I hybrids, and subsequent work on polymer-geopolymer class II hybrids is planned.) should be added as a discussion into either the introduction or the conclusion section, to make it clearer to the reader.
- I appreciate that authors modified lines 384-386, but a brief discussion of the possible reasons for the results obtained on samples 0.1wt%PBDT in comparison to the starting sample and the samples with a different loading was required.
Comments on the Quality of English LanguageThere are not particular language issues
Author Response
The authors addressed a lot of may comments. I am not fully persuaded of a couple of their answers, so I ask them to further improve the manuscript according to the following points:
- The authors reply to my concerns about the poor modifications induced by a low amount of polymer additives (We acknowledge that low amounts of polymer additives resulted in the inability to detect changes in XRD, FTIR, and EDS. However higher loadings result in viscosities that are more difficult to process. Significant changes to crystal and chemical structures are also not expected in class I hybrid materials due to a lack of strong organic-inorganic bonding interactions. The scope of this report focuses on class I hybrids, and subsequent work on polymer-geopolymer class II hybrids is planned.) should be added as a discussion into either the introduction or the conclusion section, to make it clearer to the reader.
We have included clarification regarding the scope of the report within the introduction section on lines 156 and 159-161. Additionally, a brief description regarding the processability of the PGP+Polymer resins was included in the conclusion on lines 681-682. Of note, although we did not perform additional thermal analyses outside of TGA mass loss studies, the thermo-oxidative properties of polymers at high loadings in a composite may become undesirable as large amounts of polymer degrade resulting in increased porosity or loss of mechanical reinforcement. A description of the viscosity effects on processability is also provided in lines 346-351.
The lines in the conclusion (lines 692-704) discuss in further detail (1) the inability to detect changes in XRD, FTIR, and EDS, (2) the low concentration loadings of polymer, (3) the utilization of PBDT and PBDI as Class-I geopolymer reinforcements, and (4) the future pursuit of class-II geopolymer materials.
- I appreciate that authors modified lines 384-386, but a brief discussion of the possible reasons for the results obtained on samples 0.1wt%PBDT in comparison to the starting sample and the samples with a different loading was required.
We have modified lines 384-386. This discussion has been moved to lines 390-395 to further clarify the results obtained within the 0.1 wt. % PBDT composites in comparison to the remaining resins.
Additional discussions regarding the differences in viscosity, complex viscosity, and modulus are provided in lines 403-409 and 443-449. The stark decrease in modulus for 0.1wt% PBDT is correlated with ref. 118, as commercial superplasticizers improve the workability of Portland cement through optimization of the zeta potential for electrostatic particles.
Comments on the Quality of English Language
There are not particular language issues